# *Kineococcus vitellinus* sp. nov., *Kineococcus indalonis* sp. nov. and *Kineococcus siccus* sp. nov., Isolated Nearby the Tabernas Desert (Almería, Spain)

**DOI:** 10.3390/microorganisms8101547

**Published:** 2020-10-07

**Authors:** Esther Molina-Menor, Helena Gimeno-Valero, Javier Pascual, Juli Peretó, Manuel Porcar

**Affiliations:** 1Institute for Integrative Systems Biology I2SysBio (Universitat de València-CSIC), Calle del Catedràtic Agustin Escardino Benlloch 9, 46980 Paterna, Spain; esther.molina@uv.es (E.M.-M.); juli.pereto@uv.es (J.P.); 2Darwin Bioprospecting Excellence SL. Parc Científic Universitat de València, Calle del Catedràtic Agustin Escardino Benlloch 9, 46980 Paterna, Spain; hgimeno@darwinbioprospecting.com (H.G.-V.); jpascual@darwinbioprospecting.com (J.P.); 3Departament de Bioquimica i Biologia Molecular, Universitat de València, Calle del Dr. Moliner 50, 46100 Burjassot, Spain

**Keywords:** new taxa, biocrust, Kineococcus indalonis, Kineococcus siccus, Kineococcus vitellinus, Tabernas Desert

## Abstract

Three novel Gram-positive, aerobic, chemoheterotrophic, motile, non-endospore-forming, orange-pigmented bacteria designated strains T13^T^, T90^T^ and R8^T^ were isolated from the Tabernas Desert biocrust (Almería, Spain). Cells of the three strains were coccus-shaped and occurred singly, in pairs or clusters. The three strains were oxidase-negative and catalase-positive, and showed a mesophilic, neutrophilic and non-halophilic metabolism. Based on the 16S rRNA gene sequences, the closest neighbours of strains T13^T^, T90^T^ and R8^T^ were *Kineococcus aurantiacus* IFO 15268^T^, *Kineococcus gypseus* YIM 121300^T^ and *Kineococcus radiotolerans* SRS 30216^T^ (98.5%, 97.1% and 97.9% gene sequence similarity, respectively). The genomes were sequenced, and have been deposited in the GenBank/EMBL/DDBJ databases under the accession numbers JAAALL000000000, JAAALM000000000 and JAAALN000000000, respectively, for strains T13^T^, T90^T^ and R8^T^. The average nucleotide identity (ANIb) and digital DNA-DNA hybridization (dDDH) values confirmed their adscription to three new species within the genus *Kineococcus*. The genomic G + C content of strains T13^T^, T90^T^ and R8^T^ ranged from 75.1% to 76.3%. The predominant fatty acid of all three strains was anteiso-C_15:0_. According to a polyphasic study, strains T13^T^, T90^T^ and R8^T^ are representatives of three new species in the genus *Kineococcus*, for which names *Kineococcus vitellinus* sp. nov. (type strain T13^T^ = CECT 9936^T^ = DSM 110024^T^), *Kineococcus indalonis* sp. nov. (type strain T90^T^ = CECT 9938^T^ = DSM 110026^T^) and *Kineococcus siccus* sp. nov. (type strain R8^T^ = CECT 9937^T^ = DSM 110025^T^) are proposed.

## 1. Introduction

The genus *Kineococcus,* belonging to the family *Kineosporiaceae* [1], was first proposed by Yokota et al. in 1993 [2]. At the time of writing, the genus *Kineococcus* is comprised of 11 species with valid published names (http://lpsn.dsmz.de/) which were isolated from soil [2,3,4,5], saline environments [6,7,8], plants [9,10] and radioactive work areas [11]. This genus includes species with the ability to grow in a wide range of temperatures [5] and high salt concentrations [8,9]. Their resistance to multiple stresses and extreme conditions has also been reported by several authors, with *Kineococcus radiotolerans* standing out among them because of its high resistance to radiation [11].

The genus *Kineococcus* unites Gram-stain-positive, aerobic, catalase-positive and oxidase-negative microorganisms. Colonies are cream to orange in colour, and cells are motile and occur singly, in pairs or in clusters. The major cellular fatty acid is C_15:0_ anteiso, whereas diphosphatidylglycerol and phosphatidylglycerol are the major polar lipids. The genomic DNA G + C content of the *Kineococcus* species ranges from 73.9% to 74.2% [12].

During a study on the culturable microbial diversity of biocrust in dry environments exposed to high solar radiation, three different strains, T13^T^, T90^T^ and R8^T^, were isolated. The objective of the study was to characterize in detail the microbial communities from the Tabernas Desert (Almería, Spain), with particular emphasis in the isolation of extremophilic bacteria. The bacterial strains isolated from similar environmental niches, such as solar panels or the intertidal coastal zone, are able to tolerate UV radiation, and could thus be potentially useful in the synthesis of antioxidant compounds [13,14]. Among them, pigments such as carotenoids are particularly valuable for the biotechnological and pharmaceutical industries. In this paper, we describe the isolation and polyphasic characterization of these bacteria, and propose them as new species of the genus *Kineococcus*, for which the names *Kineococcus vitellinus* sp. nov., *Kineococcus indalonis* sp. nov. and *Kineococcus siccus* sp. nov. are proposed.

## 2. Materials and Methods

### 2.1. Isolation of Strains and Culture Conditions

Biocrust samples were collected in the vicinity of the Tabernas Desert (37.002404 N, 2.450655 W) in September 2018. One gram of each biocrust sample was resuspended in one millilitre of phosphate-buffered saline (PBS) pH 7.4. The suspensions were vigorously shaken, spread on 1X, 0.1X and 0.01X trypticase soy agar (TSA; in g/L: 15 tryptone, 5 NaCl, 5 soya peptone), and Reasoner’s 2A agar (R2A; in g/L: 1 peptone, 0.5 yeast extract, 0.5 dextrose, 0.5 soluble starch, 0.3 dipotassium phosphate, 0.05 magnesium sulphate heptahydrate, 0.3 sodium pyruvate) media and incubated at 23 °C for one week. T13^T^ and T90^T^ strains were isolated from 0.1X and 0.01X TSA plates, respectively, whereas R8^T^ was isolated from a 0.1X R2A plate. All three strains were purified and cultivated by re-streaking on fresh media. For cryopreservation at −80 °C, cell suspensions in trypticase soy broth (TSB) were supplemented with 15% (*v/v*) glycerol. To determine the taxonomic status of strains T13^T^, T90^T^ and R8^T^, six reference type strains were used in this study: *Kineococcus radiotolerans* DSM 14245^T^, *Kineococcus aurantiacus* DSM 7487^T^, *Kineococcus gypseus* DSM 27627^T^ and *Kineococcus aureolus* DSM 102158^T^ from the DSMZ—German Collection of Microorganisms and Cell Cultures (Leibniz Institute, Braunschweig, Germany), and *Kineococcus gynurae* NBRC 103943^T^ and *Kineococcus mangrovi* NBRC 110933^T^ from the NITE Biological Resource Center (Tokyo, Japan). The selection of reference strains was based on the comparison of 16S rRNA gene sequences of the three isolates against the EzBioCloud database (http://www.ezbiocloud.net/) as detailed below. The comparative analysis conditions were selected according to the available literature on the species within this genus. Unless otherwise specified, all nine strains were grown on TSA media at 30 °C.

### 2.2. DNA Extraction and Sequencing

Extraction of genomic DNA was carried out using the DNeasy PowerSoil kit (QUIAGEN, Hilden, Germany) according to the manufacturer’s instructions. A PCR of the whole 16S rRNA gene was performed with universal primers 8F (5′-AGAGTTTGATCCTGGCTCAG-3′) [15] and 1492R (5′-GGTTACCTTGTTACGACTT-3′) [16] (94 °C for 5 min, 24 cycles of denaturation at 94 °C for 15 s, annealing at 48 °C for 15 s and elongation at 72 °C for 5 min, and a final elongation step at 72 °C for 5 min). The 16S rRNA gene sequence length of the strains T13^T^, T90^T^ and R8^T^ were 1415 bp (accession number MN069869), 1425 bp (MN069867) and 1404 bp (MN069868), respectively. The draft genome of strains T13^T^, T90^T^ and R8^T^ was sequenced using a MiSeq sequencer (Illumina, San Diego, CA, USA), and the Nextera XT Prep Kit protocol was used for library preparation.

### 2.3. Phylogenetic Analysis

We identified the closest relatives of the three strains T13^T^, T90^T^ and R8^T^ by comparing their 16S rRNA gene sequences against the EzBioCloud database update 2020.05.13 (http://www.ezbiocloud.net/). Maximum-likelihood (ML) (Figure 1) [17] and neighbour-joining trees (NJ) (Figure 2) [18] were inferred with the software MEGA X v.10.1.7 (https://www.megasoftware.net/). The Tamura–Nei G+I evolutionary model was used for the ML tree, whereas the Kimura two-parameter model was used for the NJ tree. The reliability of the branch patterns was assessed using bootstrap analysis based on 500 replicates for the ML tree and on 1000 replicates for the NJ tree [19]. *Acidimicrobium ferrooxidans* DSM 10331^T^ was used as an outgroup for the phylogenetic analyses.

### 2.4. Genomic Analysis

The FastQC tool [20] was used to assess the quality of the sequence reads. Genome assembly of paired reads was performed using SPAdes 3.12.0 [21]. The draft genomes were annotated using the RAST tool kit (RAStk) [22] integrated in PATRIC v.3.5.41 (https://www.patricbrc.org). The circular genomic maps were also obtained from PATRIC v.3.5.41. The whole-genome shotgun projects of strains T13^T^, T90^T^ and R8^T^ have been deposited in GenBank/EMBL/DDBJ under accession numbers JAAALL000000000, JAAALM000000000 and JAAALN000000000, respectively. The completeness and contamination of the genomes was analysed with the bioinformatic tool CheckM v.1.0.6 [23].

Pairwise average nucleotide identity values (ANIb) [24] were calculated between strains T13^T^, T90^T^ and R8^T^ and their closest type strains whose genomes were publicly available, by using the JSpeciesWS online tool [25]. Additionally, digital DNA-DNA hybridization (dDDH) pairwise values were also obtained using the Genome-to-Genome Distance Calculator 2.1 (GGDC) tool [26]. As recommended for incompletely sequenced genomes, formula 2 was used for calculating the digital DDH values [26].

### 2.5. Morphological and Biochemical Characteristics

The phenotypic characteristics of the bacterial cultures were determined after one week of growth at 30 °C on TSA medium following the procedures outlined previously by other authors [27,28]. Cell morphology was observed under the microscope with crystal violet glass stain. Motility was analysed on wet mounts. The presence of flagella was determined with the staining method described by Heimbrook et al. [29].

Growth under microaerophilic conditions was tested by incubating the plates in a candle jar. The ability to grow in anaerobiosis was determined with the BD GasPak^TM^ EZ pouch system (Becton, Dickinson and Company, Franklin Lakes, NJ, USA). Catalase activity was determined by using 30% (*v*/*v*) H_2_O_2_, recording bubble production as a positive result. Oxidase activity was determined with Oxidase Test Stick for microbiology (PanReac AppliChem, Barcelona, Spain). Gram type was determined by using KOH 3% (*w*/*v*), with cell lysis as a positive result for Gram-negative bacteria. Growth at different temperatures (4, 10, 15, 25, 30, 37, 40 and 45 °C) and at various final NaCl concentrations (0.0–4.0% at intervals of 0.5%, and 4.0–10.0% at intervals of 1.0%) was examined by cultivating the isolates on TSA medium. The ability to grow at different pH values (4.0–11.0 at intervals of 1.0 pH unit) was examined in TSB using the MES (pH 4–6), HEPES (pH 7–8) and CHES (pH 9–11) buffers at 10 mM. Carbon source assimilation and enzymatic activities were determined using the API 20NE and API ZYM system strips (bioMérieux, Marcy-l’Étoile, France) according to manufacturer’s instructions, as well as BIOLOG GEN III MicroPlates (BIOLOG Inc., Hayward, CA, USA).

### 2.6. Chemotaxonomic Analysis

For fatty acid analysis, strains T13^T^, T90^T^, R8^T^ and their closest relatives were grown on TSA at 26 °C for 72 h. Analysis of cellular fatty acids was carried out using the Sherlock Microbial Identification System (version 6.1, MIDI, Inc., Newark, DE, USA) [30]. Fatty acids were analysed on an Agilent 6859 gas chromatography system and using the MIDI method TSBA6 [31], according to the manufacturer’s instructions.

## 3. Results and Discussion

### 3.1. Phylogenetic Analysis

The almost-complete 16S rRNA gene sequences showed that the three strains were phylogenetically related to representatives of the genus *Kineococcus.* According to the EzBioCloud database tool (http://www.ezbiocloud.net/), the closest type strains of T13^T^, T90^T^ and R8^T^ are *K. aurantiacus* IFO 15268^T^ (98.5% 16S rRNA gene sequence similarity), *K. gypseus* YIM 121300^T^ (97.1%) and *K. radiotolerans* SRS 30216^T^ (97.9%), respectively. Since these 16S rRNA gene sequence similarities are lower than 98.7%, we have evidence that suggests that the strains T13^T^, T90^T^ and R8^T^ may belong to new species [32,33]. According to these results, *K. aurantiacus* DSM 7487^T^ and *K. radiotolerans* DSM 14245^T^ were selected as comparative reference strains for T13^T^; *K. gypseus* DSM 27627^T^, *K. aureolus* DSM 102158^T^ and *K. mangrovi* NBRC 110933^T^ for T90^T^; and *K. radiotolerans* DSM 14245^T^, *K. aurantiacus* DSM 7487^T^ and *K. gynurae* NBRC 103943^T^ for R8^T^.

The three isolates are classified into the genus *Kineococcus* in both 16S-rRNA-based phylogenetic trees constructed by ML and NJ. Specifically, T13^T^ showed an external position in the cluster formed by *K. aurantiacus* IFO 15268^T^*, K. mangrovi* L2-1-L1^T^, *K. endophyticus* KLBMP^T^, *K. rhizosphaerae* RP-B16^T^ and *K. radiotolerans* SRS30216^T^ (Figure 1 and Figure 2). Strain T90^T^ formed a monophyletic group with *K. gypseus* YIM 121300^T^ (Figure 1 and Figure 2) while strain R8^T^ appeared as an external taxon within the genus *Kineococcus* (Figure 1 and Figure 2). The phylogenetic inference of the three strains based on the 16S rRNA was supported by high bootstrap values.

The ANI and digital DDH values between strains T13^T^ vs. *K. radiotolerans* SRS 30216^T^ and R8^T^ vs. *K. radiotolerans* SRS 30216^T^ were 80.6–24.20% and 77.40–41.73%, respectively. As the values were higher than the threshold established to circumscribe prokaryotic species, namely 95% for ANI values [34] and 70% for dDDH [26], both genome-related indexes [35] confirmed the adscription of strains T13^T^ and R8^T^ to hitherto unknown species. On the other hand, the closest relative of T90^T^ is *K. gypseus* YIM 121300^T^. It must be noted that the genome of *K. gypseus* YIM 121300^T^ (97.1%) was not publicly available at the time of writing the manuscript. However, since the 16S rRNA gene sequence similarity between T90^T^ and its closest relative *K. gypseus* YIM 121300^T^ is <98.7%, it is not necessary to calculate any overall genome relatedness index (OGRI) to propose T90^T^ as an independent genomospecies [32].

### 3.2. Genomic Characteristics

The circular map highlighting the main genomic features of the three strains is shown in Appendix A. The draft genome of strains T13^T^, T90^T^ and R8^T^ consisted of 705, 1067 and 502 contigs, yielding a total length of 4,857,076, 4,498,067 and 4,581,425 bp, respectively. The genomic G + C content of strains T13^T^, T90^T^ and R8^T^ was 75.4%, 76.3% and 75.1%, respectively. This genomic G + C content is in accordance with other *Kineococcus* species, and further confirms their adscription to the *Kineococcus* genus [2,7,8]. A total of 5039, 4690 and 4633 coding sequences (CDSs) were predicted for strains T13^T^, T90^T^ and R8^T^, respectively, of which 1894, 2906 and 2832 were proteins with functional assignments. In the case of tRNA genes, a total of 45 were predicted for strains T13^T^ and R8^T^, and 46 were predicted for strain T90^T^. Regarding rRNA genes, six of them were identified in T13^T^; whereas five genes were found for T90^T^ and R8^T^. The genome completeness values of strains T13^T^, T90^T^ and R8^T^ were 98.9%, 98.1% and 99.2%, respectively; and the levels of contamination were 1.1%, 1.7% and 0%, respectively. Therefore, the draft genomes showed high enough quality for further analysis [32].

By analysing the draft genome of strains T13 ^T^, T90^T^ and R8^T^, their ability to synthesise diphosphatidylglycerol and phosphatidylglycerol was predicted based on the presence of genes coding for cardiolipin synthase A/B (EC:2.7.8.-) and ribosomal-protein-serine acetyltransferase (EC 2.3.1.-). However, the synthesis of phosphatidylinositol (EC 2.7.8.11), a major polar lipid of several *Kineococcus* species, could not be predicted in any of the three genomes [2,8,11]. Furthermore, the synthesis of meso-diaminopimelic acid in the three strains was predicted based on the presence of the enzyme diaminopimelate epimerase (EC 5.1.1.7), and the synthesis of menaquinones was predicted based on the identification of the enzyme demethylmenaquinone methyltransferase (EC 2.1.1.163). Therefore, the chemotaxonomic profile of the three new strains is in accordance with other *Kineococcus* species, corroborating their inclusion into the genus *Kineococcus* [2,7,8].

### 3.3. Phenotypic Characterization

Strains T13^T^, T90^T^ and R8^T^ proved to be aerobic, Gram-positive, motile and coccus-shaped (1 µm in diameter). Like other members of the genus *Kineococcus*, cells of the three strains occur singly, in pairs or in clusters (Appendix A). Cell motility was confirmed by the presence of flagella. Colonies were orange-coloured, with irregular margins and a rough surface. T13^T^ colonies were paler than other *Kineococcus* strains. T90^T^ colonies changed in colour to dark orange-greenish after incubation at low temperatures (below 15 °C). The colonies displayed a diameter of around 1–2 mm after 3–4 days of growth, although T90^T^ colonies displayed a larger diameter.

All three strains were able to grow between 10 and 30 °C (optimum 25–30 °C). Furthermore, strain R8^T^ was able to grow at 4 °C, while strains T13^T^ and T90^T^ could grow up to 37 and 40 °C, respectively (Table 1). Strains T90^T^ and R8^T^ could grow at NaCl concentrations up to 4%, while T13^T^ could grow at concentrations up to 5% (optimum for the three strains, 0% NaCl). *K. radiotolerans* DSM 14245^T^, *K. aurantiacus* DSM 7487^T^, *K. gypseus* DSM 27627^T^ and *K. gynurae* NBRC 103943^T^ showed a salinity tolerance range similar to our strains, while *K. aureolus* DSM 102158^T^ and *K. mangrovi* NBRC 110933^T^ were able to grow at concentrations up to 8% and 9% NaCl, respectively (Table 1). Even though the three strains were found to be neutrophilic (optimum growth at pH 7), strains T13^T^ and T90^T^ were able to grow at alkaline values (up to pH 9 and 10, respectively), whereas R8^T^ could grow weakly at pH 5. Other *Kineococcus* species such as *K. aureolus* DSM 102158^T^, *K. aurantiacus* DSM 7487^T^, *K. gypseu*s DSM 27627^T^, *K. mangrovi* NBRC 110933^T^ and *K. gynurae* NBRC 103943^T^ were able to tolerate alkaline pH (Table 1). All three strains were able to grow under microaerophilic conditions, but no growth was observed for any of them in anaerobiosis.

Strains T13^T^, T90^T^ and R8^T^, like their closest relatives, showed a positive response for esculin hydrolysis, *β*-galactosidase, esterase (C4), esterase lipase (C8), leucine arylamidase, α-chymotrypsin, acid phosphatase, naphthol-AS-BI-phosphohydrolase and *β*-glucosidase. On the other hand, strains T13^T^, T90^T^ and R8^T^ and their closest neighbours showed a negative response for nitrate reduction, indole production, glucose fermentation, arginine dihydrolysis and N-acetyl-*β*-glucosaminidase. Strains R8^T^ and *K. gynurae* NBRC 103943^T^ showed urease activity, while strain T90^T^, *K. gynurae* NBRC 103943^T^, *K. aureolus* DSM 102158^T^ and *K. gypseus* DSM 27627^T^ were able to hydrolyse gelatine (Table 1). In API 20 NE strips, strain R8^T^ proved able to use all the saccharides tested (Table 1). On the contrary, strain T13^T^ was only able to grow weakly with glucose, while T90^T^ could not grow in any of the carbon sources tested. Furthermore, BIOLOG assays revealed that strains T13^T^, T90^T^ and R8^T^ were able to oxidize 31, 62 and 26 out of the 71 tested carbon sources, respectively, and these were mainly organic acids and amino acids (Appendix A). This suggests that strain T90^T^ displays a more polytrophic metabolism than strains T13^T^ and R8^T^.

The major fatty acids for strains T13^T^, T90^T^ and R8^T^ were anteiso-C_15:0_ (96.1%, 73.9% and 94.0%, respectively). Similarly, their closest relatives showed high amounts of anteiso-C_15:0_ (Table 2), corroborating the inclusion of the three strains in the genus *Kineococcus* [2,8,11]. The presence of anteiso-C_15:0_ fatty acid in the cell membranes has been associated with resistance to low temperatures [36,37]. In fact, this fatty acid has been found in the psychrophilic members of the genus *Flavobacterium* [38] isolated from a cold desert environment. Moreover, anteiso-C_15:0_ fatty acid can play an important role in biofilm formation [39,40], which is in accordance with the biofilms and cell clumps observed for the three *Kineococcus* strains under liquid growth conditions. 

Unlike their closest relatives, strains T90^T^ and *K. gypseus* DSM 27627^T^ also showed high amounts of anteiso-C_15:1_ (16.9%) (Table 2). Furthermore, the presence or absence of other minor fatty acids allowed us to differentiate the new strains from their closest neighbours (Table 2).

## 4. Conclusions

During a study on the microbial communities associated with biocrust in sun-exposed and dry surfaces, the three strains T13^T^, T90^T^ and R8^T^ were isolated in pure culture. All three strains, which were phylogenetically related to the genus *Kineococcus*, were characterized using a polyphasic approach. According to the phenotypic, genomic and phylogenetic analyses, strains T13^T^, T90^T^ and R8^T^ should be considered as new species within the *Kineococcus* genus, for which the names *Kineococcus vitellinus* sp. nov., *Kineococcus indalonis* sp. nov. and *Kineococcus siccus* sp. nov., respectively, are proposed.

### 4.1. Description of Kineococcus vitellinus sp. nov.

#### *Kineococcus vitellinus* (vi.tel.li’nus. N.L. masc. adj. Vitellinus Egg-Yolk-Coloured)

Cells are Gram-positive, motile, non-endospore-forming, catalase-positive, oxidase-negative cocci (1 µm in diameter). Cells occur singly, in pairs or in clusters. Colonies are 1–1.5 mm in diameter, circular, rough and pale orange. Temperature range for growth is 10–37 °C, with an optimum at 25–30 °C. Growth occurs at pH 6–9 (optimum pH 7.0) and tolerates up to 5% NaCl (*w*/*v*), with optimum at 0% NaCl (*w*/*v*). Esculin hydrolysis, β-galactosidase, esterase (C4), esterase lipase (C8), leucine arylamidase, α-chymotrypsin, acid phosphatase, naphthol-AS-BI-phosphohydrolase, β-glucosidase, valine arylamidase, α-galactosidase, α-glucosidase, α-mannosidase and α- fucosidase activities are detected. Nitrate reduction, indole production, glucose fermentation, arginine dihydrolysis, urease, gelatinase, alkaline phosphatase, lipase (C14), cystine arylamidase, trypsin, β-glucuronidase and N-acetyl-β-glucosaminidase are not detected. Using API 20NE test kit, this species is weakly positive for the assimilation of glucose and negative for arabinose, mannitol, N-acetyl-glucosamine, maltose, potassium gluconate, malic acid, mannose, capric acid, adipic acid, trisodium citrate and phenylacetic acid. Using BIOLOG GENIII MicroPlates, this species is positive for the utilization of d-raffinose, α- d-glucose, d-sorbitol, pectin, Tween 40, dextrin, α-d-lactose, d-mannose, d-mannitol, d-maltose, d-melibiose, d-fructose, l-alanine, d-trehalose, β-methyl-d-glucoside, d-galactose, myo-inositol, d-cellobiose, d-salicin, glycerol, gentiobiose, N-acetyl-d-glucosamine, glucuronamide, acetoacetic acid, sucrose, d-fructose-6PO_4_, d -turanose, l-rhamnose, l-pyroglutamic acid, stachyose and l-serine; and negative for the utilization of gelatine, p-hydroxy-phenylacetic acid, glycyl- l-proline, d-galacturonic acid, methyl pyruvate, γ-amino-butyric acid, d-arabitol, l-galactonic acid lactone, d-lactic acid methyl ester, α-hydroxy-butyric acid, l-arginine, d-gluconic acid, l-lactic acid, β-hydroxy- d, l-butyric acid, 3-methyl- d-glucoside, l-aspartic acid, d-glucuronic acid, citric acid, α-keto-butyric acid, d-fucose, d -glucose-6-PO_4_, l-glutamic acid, β-keto-glutaric acid, N-acetyl-β-d-mannosamine, l-fucose, l-histidine, mucic acid, d-malic acid, propionic acid, N-acetyl-d-galactosamine, d-aspartic acid, quinic acid, l-malic acid, acetic acid, N-acetyl neuraminic acid, inosine, d-serine, d-saccharic acid, bromo-succinic acid and formic acid. The major fatty acid is anteiso-C_15:0_. The type strain T13^T^ (CECT 9936^T^ = DSM 110024^T^) was isolated nearby the Tabernas Desert in Almería (Spain), from biocrust samples. The G + C content of the type strain is 75.4%. The GenBank/EMBL/DDBJ accession number for the 16S rRNA sequence is MN069869, and the genome accession number is JAAALL000000000.

### 4.2. Description of Kineococcus indalonis sp. nov.

#### *Kineococcus indalonis* (in.da.lo’nis. N.L. gen. n. *indalonis* of Indalo, Which Is a Prehistoric Symbol Found in Rock Paintings in Almería (Spain), Referring to the Place Where the Microorganism Was Isolated)

Cells are Gram-positive, motile, non-endospore-forming, catalase-positive, oxidase-negative cocci (1 µm in diameter). Cells occur singly, in pairs or in clusters. Colonies are small (1 mm in diameter), circular with irregular margins, rough and pale orange, but below 20 °C the colour changes from orange to dark greenish. Temperature range for growth is 10–40 °C, with an optimum at 25–30 °C. No growth is observed below 10 or above 40 °C. Growth occurs at pH 6–10 (optimum, 6–9) and tolerates up to 4% NaCl (*w*/*v*), with optimum at 0% (*w*/*v*). Esculin hydrolysis, β-galactosidase, gelatinase, esterase (C4), esterase lipase (C8), leucine arylamidase, α-chymotrypsin, acid phosphatase, alkaline phosphatase, valine arylamidase, α-galactosidase, naphthol-AS-BI-phosphohydrolase, α-glucosidase, α-mannosidase and β-glucosidase activities are detected. Nitrate reduction, indole production, glucose fermentation, arginine dihydrolysis, urease, lipase (C14), cystine arylamidase, trypsin, β-glucuronidase, α-fucosidase and N-acetyl-β-glucosaminidase are not detected. Using API 20NE test kit, this species is negative for the assimilation of glucose, arabinose, mannitol, N-acetyl-glucosamine, maltose, potassium gluconate, malic acid, mannose, capric acid, adipic acid, trisodium citrate and phenylacetic acid. Using BIOLOG GENIII MicroPlates, this species is positive for the utilization of d-raffinose, α-d-glucose, d-sorbitol, gelatine, pectin, Tween 40, dextrin, α- d-lactose, d-mannose, d-mannitol, glycyl-l-proline, d-galacturonic acid, γ-amino-butyric acid, d-maltose, d-melibiose, d-fructose, d-arabitol, l-alanine, l-galactonic acid lactone, d-lactic acid methyl ester, d-trehalose, β-methyl-d-glucoside, d-galactose, myo-inositol, l-arginine, d-gluconic acid, β-hydroxy-d,l-butyric acid, d-cellobiose, d-salicin, 3-methyl- d-glucoside, glycerol, l-aspartic acid, d-glucuronic acid, citric acid, α-keto-butyric acid, gentiobiose, N-acetyl-d-glucosamine, d-fucose, d-glucose-6-PO_4_, l-glutamic acid, glucuronamide, α-keto-glutaric acid, acetoacetic acid, sucrose, N-acetyl-β-d-mannose, l-fucose, d-fructose-6PO_4_, l-histidine, mucic acid, d-malic acid, propionic acid, d-turanose, N-acetyl-d-galactosamine, l-rhamnose, d-aspartic acid, l-pyroglutamic acid, l-malic acid, acetic acid, stachyose, d-serine, l-serine and bromo-succinic acid; and negative for the utilization of p-hydroxy-phenylacetic acid, methyl pyruvate, α-hydroxy-butyric acid, l-lactic acid, quinic acid, N-acetyl neuraminic acid, inosine, d-saccharic acid and formic acid. The major fatty acids for strain T90^T^ are anteiso-C_15:0_ and anteiso-C_15:1_ A. The type strain T90^T^ (CECT 9938^T^ = DSM 110026^T^) was first isolated nearby the Tabernas Desert in Almería (Spain), from biocrust samples. The G + C content of the type strain is 76.3%. The GenBank/EMBL/DDBJ accession number for the 16S rRNA sequence is MN069867, and the genome accession number is JAAALM000000000.

### 4.3. Description of Kineococcus siccus sp. nov.

#### *Kineococcus siccus* (*siccus*. L. masc. adj. *Siccus*, Dry)

Cells are Gram-positive, motile, non-endospore-forming, catalase-positive, oxidase-negative cocci (1 µm in diameter). Cells occur singly, in pairs or in clusters. Colonies are orange, circular, with irregular margins and variable size (1–2 mm diameter). Temperature range for growth is 4–30 °C with an optimum at 25–30 °C. No growth is observed at 40 °C. Growth occurs at pH 5–8 (optimum 6–7) and tolerates up to 4% NaCl (*w*/*v*), with optimum at 0% (*w*/*v*). Esculin hydrolysis, β-galactosidase, urease, esterase (C4), esterase lipase (C8), leucine arylamidase, α-chymotrypsin, acid phosphatase, alkaline phosphatase, valine arylamidase, cystine arylamidase, α-galactosidase, naphthol-AS-BI-phosphohydrolase, α-fucosidase and β-glucosidase activities are detected. Nitrate reduction, indole production, glucose fermentation, arginine dihydrolysis, gelatinase, lipase (C14), trypsin, α-glucosidase, β-glucuronidase, α-mannosidase and N-acetyl-β-glucosaminidase are not detected. Using API 20NE test kit, this species is positive for the assimilation of glucose, arabinose, mannose, mannitol, N-acetyl-glucosamine and maltose; weak for the assimilation potassium gluconate and adipic acid and negative for the assimilation of malic acid, capric acid, trisodium citrate and phenylacetic acid. Using BIOLOG GENIII MicroPlates, this species is positive for the utilization of pectin, Tween 40, dextrin, d-fructose, d-arabitol, l-alanine, d-galactose, d-gluconic acid, l-lactic acid, d-cellobiose, 3-methyl-d-glucoside, glycerol, d-glucuronic acid, gentiobiose, glucuronamide, acetoacetic acid, l-fucose, d-fructose-6-PO_4_, mucic acid, d-malic acid, d-turanose, d-aspartic acid, quinic acid, l-malic acid, acetic acid and d-serine; and negative for the utilization of d-raffinose, α-d-glucose, d-sorbitol, gelatine, p-hydroxy-phenylacetic acid, α-d-lactose, d-mannose, d-mannitol, glycyl-l-proline, d-galacturonic acid, methyl pyruvate, γ-amino-butyric acid, d-maltose, d-melibiose, l-galactonic acid lactone, d-lactic acid methyl ester, α-hydroxy-butyric acid, d-trehalose, β-methyl-d-glucoside, myo-inositol, l-arginine, β-hydroxy-d,l-butyric acid, d-salicin, l-aspartic acid, citric acid, α-keto-butyric acid, N-acetyl-d-glucosamine, d-fucose, d-glucose-6-PO_4_, l-glutamic acid, α-keto-glutaric acid, sucrose, N-acetyl-β-d-mannose, l-histidine, propionic acid, N-acetyl-d-galactosamine, l-rhamnose, l-pyroglutamic acid, stachyose, N-acetyl neuraminic acid, inosine, l-serine, d-saccharic acid, bromo-succinic acid and formic acid. The major fatty acid is anteiso-C_15:0_. The type strain R8^T^ (CECT 9937^T^ = DSM 110025^T^) was first isolated nearby the Tabernas Desert in Almería (Spain), from biocrust samples. The G + C content of the type strain is 75.1%. The GenBank/EMBL/DDBJ accession number for the 16S rRNA sequence is MN069868, and the genome accession number is JAAALN000000000.

## Figures and Tables

**Figure 1 microorganisms-08-01547-f001:**
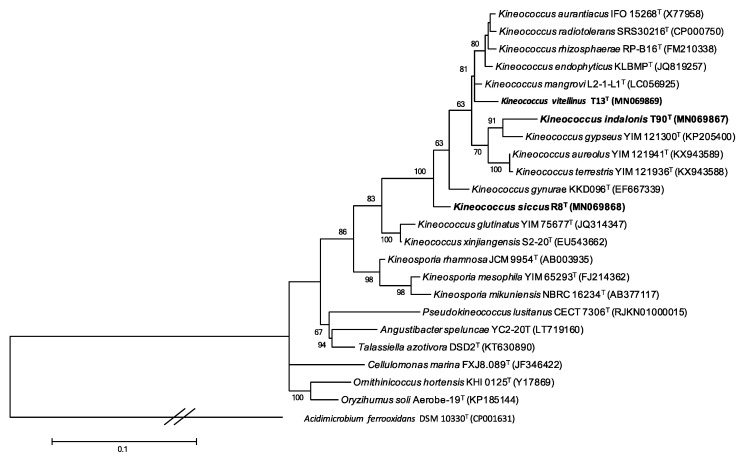
Maximum likelihood phylogenetic tree showing the relationships between strains T13^T^, T90^T^, R8^T^ and other members of the family *Kineosporiaceae* based on 16S rRNA sequences. The optimal evolutionary model of nucleotide substitution applied was Tamura–Nei G+I. Numbers at branch points refer to bootstrap percentages based on 500 replicates (values under 50% are not indicated). *Acidimicrobium ferrooxidans* DSM 10331^T^ (CP001631) was used as an outgroup. Bar 0.1 fixed nucleotide substitutions per site.

**Figure 2 microorganisms-08-01547-f002:**
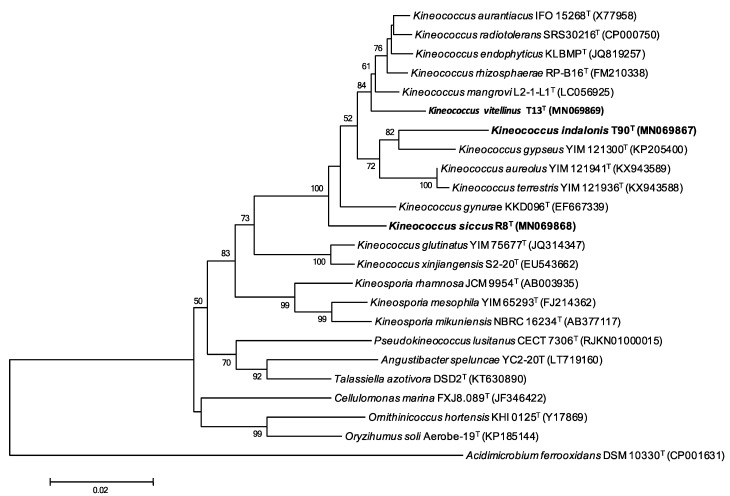
Neighbour-joining phylogenetic tree showing the relationships between strains T13^T^, T90^T^, R8^T^ and other members of the family *Kineosporiaceae* based on 16S rRNA sequences. The evolutionary model of nucleotide substitution applied was the Kimura two-parameter model (K2P). Numbers at branch points refer to bootstrap percentages based on 1000 replicates (values under 50% are not indicated). *Acidimicrobium ferrooxidans* DSM 10331^T^ (CP001631) was used as an outgroup. Bar 0.02 fixed nucleotide substitutions per site.

**Table 1 microorganisms-08-01547-t001:** Differential characteristics between strains T13^T^, T90^T^, R8^T^ and related type strains of the genus *Kineococcus*.

Characteristic	1	2	3	4	5	6	7	8	9
Isolation source	Biocrust	Biocrust	Biocrust	Radioactive work area [11]	Saline sediment [8]	Soil [1]	Saline sediment [6]	Mangrove sediment [7]	Medicinal plant [9]
**Growth at/in:**									
Temperature range (°C)	10–37	10–40	4–30	4–37	10–45	4–40	15–37	10–30	4–30
pH range	6–9	6–10	5–8	6–8	7–11	5–9	5–10	5–9	5–10
NaCl tolerance (%, *w*/*v*)	0–5	0–4	0–4	0–4	0–8	0–4	0–4	0–9	0–6
**Carbon source utilization** **(API 20NE)**									
d-Glucose	W	-	+	+	+	+	+	+	+
l-Arabinose	-	-	+	+	+	+	+	+	+
d-Mannose	-	-	+	+	+	+	+	+	+
d-Mannitol	-	-	+	+	+	+	W	+	+
N-Acetyl-glucosamine	-	-	+	W	-	+	-	W	-
d-Maltose	-	-	+	W	+	+	W	+	+
Potassium gluconate	-	-	W	+	+	W	-	+	W
Adipic acid	-	-	W	W	W	-	-	-	-
Malic acid	-	-	-	W	+	-	W	+	W
Trisodium citrate	-	-	-	-	W	-	-	-	W
Phenylacetic acid	-	-	-	-	W	-	-	-	-
**Enzymatic activity (API 20NE)**									
Urease	-	-	+	-	-	-	-	-	+
Gelatinase	-	+	-	-	+	**-**	+	-	+
**Enzymatic activity (API ZYM)**									
Alkaline phosphatase	-	+	+	+	-	+	-	+	+
Lipase (C14)	-	-	-	+	-	-	-	-	-
Valine arylamidase	+	+	+	+	+	+	+	-	-
Cystine arylamidase	-	-	+	+	-	+	+	-	-
Trypsin	-	-	-	+	-	-	-	-	-
*α*-Galactosidase	+	+	+	+	+	+	-	+	+
*β*-Glucuronidase	-	-	-	+	-	-	-	-	-
*α*-Glucosidase	+	+	-	+	-	-	+	+	+
*α*-Manosidase	+	+	-	-	+	+	-	-	+
*α*-Fucosidase	+	-	+	-	+	-	-	-	-

Strains: 1, T13^T^; 2, T90^T^; 3, R8^T^; 4, *Kineococcus radiotolerans* DSM 14245^T^; 5, *Kineococcus aureolus* DSM 102158^T^; 6, *Kineococcus aurantiacus* DSM 7487^T^; 7, *Kineococcus gypseus* DSM 27627^T^; 8, *Kineococcus mangrovi* NBRC 110933^T^; 9, *Kineococcus gynurae* NBRC 103943^T^. Data in the present study were obtained from experiments carried out under identical conditions. All strains were positive for the following characteristics: Gram reaction, catalase activity, esculin hydrolysis, *β*-galactosidase, esterase (C4), esterase lipase (C8), leucine arylamidase, *α*-chymotrypsin, acid phosphatase, naphthol-AS-BI-phosphohydrolase and *β*-glucosidase. All strains were negative for the following characteristics: oxidase, nitrate reduction, indole formation, glucose fermentation, arginine dihydrolysis, capric acid assimilation and N-acetyl-*β*-glucosaminidase. +, positive; -, negative; W, weak reaction.

**Table 2 microorganisms-08-01547-t002:** Cellular fatty acid composition of strains T13^T^, T90^T^ and R8^T^ and their closest relatives.

Fatty Acids	1	2	3	4	5	6	7	8	9
**Saturated**									
C_14:0_	tr	1.2	1.37	-	1.8	1.3	2.1	3.9	1.6
iso-C_14:0_	-	1.7	1.34	-	2.0		1.8	7.1	2.9
C_15:0_	-	tr	-	-	1.7	1.0	2.6	-	2.2
iso-C_15:0_	tr	1.3	-	-	tr	-	-	-	-
anteiso-C_15:0_	96.1	73.9	94.0	97.9	78.8	75.9	73.0	84.4	74.2
C_16:0_	-	tr	tr	-	2.0	1.8	2.2	tr	1.8
C_18:0_	-	-	tr	-	tr	1.5	1.2	-	-
**Unsaturated**									
AT 12–13 C_13:1_	-	-	-	-	-	3.4	-	-	3.4
anteiso-C_15:1_ A	2.6	16.9	-	tr	6.7	-	12.3	-	-
**Hydroxylated**									
C_14:0_ 2-OH	-	-	-	-	-	5.4	-	tr	4.0
C_17:0_ 2-OH	-	-	-	-	-	-	-	2.8	-
C_17:0_ 3-OH	-	3.4	tr	1.2	3	2.4	3.5	-	1.3
**Summed ***									
Sum in feature 2	-	-	-	-	-	2.7	-	-	2.8
Sum in feature 3	-	-	-	-	-	2.7	-	-	2.4
Sum in feature 5	-	-	-	-	-	2.0	-	-	-

* Summed features represent groups of fatty acids that cannot be separated with the chromatographic system. Sum in feature 2 corresponds to C_13:0_ 3-OH/iso-C_15:1_ I/iso-C_15:1_ H. Sum in feature 3 corresponds to iso-C_16:1_ I/C_14:0_ 3-OH/C_12:0_ alde? Sum in feature 5 corresponds to iso-C_17:1_ I/anteiso-C_17:1_ B.; Strains: 1, T13^T^; 2, T90^T^; 3, R8^T^; 4, *Kineococcus radiotolerans* DSM 14245^T^; 5, *Kineococcus aureolus* DSM 102158^T^; 6, *Kineococcus aurantiacus* DSM 7487^T^; 7, *Kineococcus gypseus* DSM 27627^T^; 8, *Kineococcus mangrovi* NBRC 110933^T^; 9, *Kineococcus gynurae* NBRC 103943^T^. Data for reference strains were obtained in the present study. Values are shown as percentages. – means not detected; tr, <1.0% trace.

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
