# Peer review of "Kineococcus vitellinus sp. nov., Kineococcus indalonis sp. nov. and Kineococcus siccus sp. nov., Isolated Nearby the Tabernas Desert (Almería, Spain)"

_microorganisms, 2020, doi:10.3390/microorganisms8101547_

Round 1
Reviewer 1 Report
L45: 'Gram'.
L162: '... the closest type strainS ... are ...'; also missing a 'respectively' at the end.
L166: remove 'a' in 'belong to a new species' - gives the idea that all three isolates belong to the same new species.
L214: 'However, since the 16S rRNA gene sequence similarity between ...'
L366: missing a period at the end of the sentence.
Reviewer 2 Report
On the whole, the matter of the report is in general agreement with the topic of the journal "Microorganisms". The purpose of the article is to isolate and describe three new species of the bacteria Kineococcus, which were sampled from the biocrust of Tabernas desert (Almería, Spain). However, the relevance of the study was not presented in the best way, although the report is original, the authors’ conclusions and the results may be of interest to microbiologists. The structure of the presentation of the results requires significant improvement and supplementation with illustrations. This could make the report more understandable and more attractive to the perception. Many corrections and clarifications are required.
1) Abstract does not reflect the main achievements of the study. Abstract should be revised and added by the information that the draft genome and the full 16S rRNA of strains T13T, T90T and R8T were sequenced.
2) From the Introduction it is not clear why this research is important. Why did the authors isolate these strains? Do the isolated strains have some important characteristics? Can the strains be of use in industry or in some other sphere of national economy? This should be described more completely.
3) Results:
50-51: The authors specify that "During a study on the microbial diversity of biocrust in dry and highly irradiated environments, three different strains, T13T, T90T, and R8T, were isolated". If it was a radioactive environment, this should be specified more precisely. What type of radioactive radiation affected the strains?
329: It is not clear what the authors mean when they write "highly irradiated". Were it the conditions of high radioactivity exposure? If it was radioactivity exposure, then the authors should clarify the type and some characteristics of the radioactivity so that readers can more accurately understand what was meant.
230: It would be nice to present circular maps of the three new Kineococcus sequenced genomes with genome features in the Result section.
286: It would be more revealing if the authors presented curves of growth versus pH, T and salt concentration, instead of listing incomplete information on these properties in Table-1.
313: The strains have a high anteiso-C15:0 content. What are the authors' assumptions about the benefits of this phenotype? Are there any useful properties from these compounds?
313-314: It seems that in the following statement the authors made a slip of a pen and there should be "high amounts of anteiso-C15:0 (Table 2)" instead of "high amounts of anteiso-C15:1 (Table 2)".
138-139: The authors write that the strains are oxidase negative when they tested strans for oxidase activity with Oxidase Test Stick for microbiology (PanReac AppliChem, Barcelona, Spain). However, the sequenced genomes of the strains show that they contain genes of the typical aa3-type cytochrome oxidase. This means that there is a possibility that the strains are oxidase-positive, and the tests used for some reason do not give an objective picture. For this reason, it is necessary in the description of the method where the test with strips for oxidase is used, briefly specify the principles of the test used. In addition, I suggest to carry out an inexpensive and simple additional test: treat colonies with the N,N,N ',N'-tetramethyl-p-phenylenediamine dihydrochloride (TMPD) solution at 37 °C to evaluate the capability of the strains for oxidase reaction. And it is reasonable to indicate in the description of the strains that they have genes for typical cytochrome oxidase and give the results of tests with strips and with TMPD.
Some sentences should be corrected throughout the text, taking into consideration the following notations: [...] for inclusion and ]...[ for deletion:
5: Comma was omitted before Manuel: "Juli Peretó1,2,3 [,] Manuel Porcar"
45: G[r]am-stain-positive
45-46: The genus Kineococcus harbours?
[The genus Kineococcus unites gram-positive, aerobic, catalase-positive and oxidase-negative microorganisms.]
46-47: ]Colonies are cream coloured to orange[
[Colonies are cream to orange in color ...]
92: We identified the closest relatives of the three strains [T13T, T90T, R8T]
144: What were the concentrations of the used buffers? It should be indicated in Methods
163: the closest type strain[s] of T13T, T90T and R8T are K. aurantiacus IFO 15268T ...
214-215: between T90T and ]its closest relative[ [K. gypseus YIM 121300T] is < 98.7%
215: it is not necessary [to] calculate
240: the three new strains is in accordance [with] other Kineococcus species
276: could grow at [concentrations] up to 5 %
279: were able to grow at [concentrations] up to 8
281: at ]alkalophilic[ [alkaline] values
305: were able to ]hydrolysate[ [hydrolyze] gelatin
306-307: There is no need to enumerate all the saccharides in the text as they are listed in Table 1. Suffice it to say that "strain R8T [was able to use all the saccharides tested (Table 1)]..."
Round 2
Reviewer 2 Report
No suggestions
This manuscript is a resubmission of an earlier submission. The following is a list of the peer review reports and author responses from that submission.
Round 1
Reviewer 1 Report
The manuscript is very well written and the work is well exposed for the reader.
Isolates were thoroughly tested in a polyphasic approach with several different methods to reach the conclusion that these three strains makeup three novel species. However, some important genome-based comparisons are missing.
General comment: As is, sadly, observed in many novel species papers, the novel species are described having only one strain at study. It would be ideal to have more strains when describing a novel species to better understand the spectra of its physiology, genomic variety and other traits. However, it is understandable that this is not possible with every study.
Introduction:
L40: 'high' instead of 'highly'.
Materials&methods:
L58: 'R8T, six' instead of 'R8T. Six'.
L70: space missing between 'T90T' and 'and'.
L106-111: predictions based on genes found in genome sequencing. I expect the authors to keep extrapolation from these informations to a minimum and/or expose a solid discussion on how inferring/predicting function from genes does not assure that these genes are expressed and under which conditions (in the region L203-233).
L114: did the authors only use agar in this medium for growth of the isolates? Or was it TSA/R2A depending on the isolate? Or another medium?
Results&discussion:
L147: Although there are new guidelines available for the cutoff for what is considered a novel species based on the 16S sequence, I would suggest the authors to change their statement from 'we can *confirm* strains ... belong to a new species' to something like 'we have evidence that suggests that these strains may belong to novel species'. In fact, if 16S seq would suffice to 'confirm' the status of novel species, no other tests tests would be required (eg ANIb or dDDH or GGDC).
L152: The authors analyze the phylogenetic tree critically, however I would like to suggest the addition of a phrase to emphasize that this result shows that all three strains are well integrated in the Kineococcus genus.
L159-169: In both legends, there is no mention of the scale presented in the figures. Please add the meaning of the scale to the legends. On a related note, the legend for nucleotide substitution rate in Figure 3 (L180-181) does not match the scale that is presented in the figure.
L170-175: According to the authors, building and analyzing the phylogenomic tree would allow them to obtain a more accurate phylogenetic infere. According to Figure 3, only two of the six type strains that the authors found to be the most closely related to the strains at study, were included in the phylogenomic tree analysis. Thus, all of the three closest relatives to strain T90T (K. gypseus, K aureolus and K mangrovi) and one of three closest relative of strain R8T (K gynurae) are not included in this analysis. This lack of comparison material diminishes the informative potential of the phylogenomic analysis. So, considering that (i) Phylogenetic inference based on near-complete 16S gene sequence suffices for comparisons in this genus (unlike some genera that require multi loci analysis to obtain a correct phylogenetic inference); and (ii) that not all information necessary for an accurate analysis of phylogenomic tree is available for use, authors should consider either (i) sequencing the genomes of the closest relatives that are missing and include those in the analysis, or (ii) remove the analysis from the manuscript. As is, it is not a full picture and it does not offer any additional information that cannot be gathered from phylogenetic analyses of the near-complete 16S.
L182-187: Same issue as above. Authors perform ANIb/dDDH analyses on the novel strains versus publicly available genomes, which means comparing to adequate type species (T13/R8 vs K aurantiacus and K radiotolerans); not comparing relevant type species (T90 to all three closest neighbours!; R8 vs K gynurae); and comparing against type species that have available genomes but are not the most relevant according to phylogenetic proximity to the novel strains. Again, this renders the analysis incomplete to say the least. Now, considering that near-complete 16S gene seq comparinson showed high values for the comparison of all novel strains against previsouly described species of Kineococcus, and also considering that when this is the case, there should be a genome-based analysis to confirm if novel strains are in fact a new species or not (please re-consult reference 20: Chun et al 2018 for Proposed minimal standards for the use of genome data for the taxonomy of prokaryotes, doi: 10.1099/ijsem.0.002516), authors can only have a genome-based confirmation of species novelty when the comparisons are made to the relevant type species.
L188: Table 1 should be re-formatted to better fit the values and brackets in a single line.
L275-276: Table 2 instead of Table 1.
L283, 286, 287, 288: Table 3 instead of Table 2.
Also attached the supplementary file with only 2 typo corrections.

Reviewer 2 Report
This paper proposes three new species of the genus Kineococcus. Proposing three new species required detailed data. However, this study lacks appropriate comparative analysis with the closely related taxa. Furthermore, the descriptions of the methods are inadequate. The paper does not actually follow all the norms of taxonomic guidelines. I am hesitating to accept this manuscript at this stage. The taxonomic description of the member of this genus is valuable. So, I would like to give you the opportunity to revise the manuscript. The author should not ignore any of the suggested analysis or suggestion. Following the comments below to revise this manuscript could improve the paper highly:
Comments:
- Line 35: Please update the member of the genus Kineococcus by referring to either the LPSN database or Names for Life Database.
- The introduction section should be elaborated by including the chemotaxonomic features, GC mol%, physiological and biochemical properties of the Genus. Any environmental or other potentially applicable features of this genus should also be included.
- Line 56: replace the word “came” with “isolated”
- Line 58: The sentence “To determine the taxonomic status of strains T13T, T90T, and R8T.” seems incomplete. Please rewrite this sentence.
- Section 2.1, Line 58-62: On what basis the reference strains were selected? This should be stated appropriately. Actually Lines 147-151 should be transferred to the methodology section.
- Please provide the condition for the storage and maintenance of the strains. This is essential for the readers as if they want to use the strains in the future.
- In section 2.2, several references are missing. The author should acknowledge the previous work of the author by citing the reference. For example: provide the reference for primer used for 16S rRNA, the algorithm used for generating phylogenetic tree. These are only some of the example which the author failed to cite the appropriate reference. There are several sections where the author must include appropriate references.
- Which server or which database was used to compare the 16S rRNA gene sequence. This should be mentioned in section 2.2
- Please provide the data for the hydrolysis test and some commercial media tests. This information helps the researcher for the potential use of this strain appropriately.
- The chemotaxonomic data should be further supported by Polar lipid, Quinone, Polyamine, Peptidoglycan, and whole-cell sugar analysis. These are basic features that must be provided for gram-positive bacteria and are required to describe chemotaxonomic properties precisely. Although the author has performed genome mining for these properties, the genome basis for these properties must be ascertained by conducting these experiments in the lab.
- Please do not hesitate to provide appropriate references for the methodology in section 2.4 (Morphological and Biochemical characteristics)
- Line 145-146: The comparison of 16S rRNA must be rechecked, as the data do not reveal exactly what the author stated in the manuscript.
- The morphology features should be depicted by TEM or SEM analysis. Please provide TEM or SEM images for all three strains.
- Also, the author should be assessed for the presence of flagella by TEM or SEM analysis.
- Line 153-154: Provide the strain no. for the illustrated strains. Also please mentioned clearly whether these strains are Type strains or not. If these are type strains, the “T” should be mentioned in the superscript after strain number.
- In addition to ANI/dDDH, it is also strongly suggested to provide the AAI data.
- As the author has the genome data, the author should compare the genome of the novel species with the closest reference strains. Comparative genomic features should be illustrated in the table and must be provided in the main text.
- Is there any specific features of the genome of these proposed strains?
- As the genome analysis revealed the presence of carotenoid, the author should analyze the presence of carotenoid by using a spectrophotometer and provide the peak image of the carotenoid pigment in the supplementary file.
- BOX –PCR or rep-PCR should be performed to illustrate the difference in the DNA fingerprinting.
- The abstract should be elaborated by adding some genomic properties, ANI, dDDH, and GC mol%.
- The English language should be improved throughout the manuscript. Some typo errors, grammatical errors, and long sentences existed which can mislead the readers.
Round 2
Reviewer 2 Report
The author has not attempted to follow most of the suggestions provided in the previous review. Based on only genetic analysis, it could not seem appropriate to established these taxa as novel species. Most of the genetic features might remain silent without expressing. So, at the study time, the phenotypic, biochemical, chemotaxonomy, and other properties are essential for the readers. Even, the morphology features based on only primitive optical data is insufficient. Any new taxon description should accompany with TEM or SEM images. To propose at least a novel taxon, the author must consider basic norms as prescribed by the official Journal IJSEM.
Next, what is the main specialty of these novel species? There are numerous novel taxa that prevail in the surroundings. Without any functional behavior, proposing only novel taxa based on very primitive analysis does not fit for highly competitive journals like “Microorganisms. So, this reviewer strongly rejects this paper and suggest authors add minimum analysis and mine the genome to reveal the specialty of these taxa.